# Effects of Oak Leaf Extract, Biofertilizer, and Soil Containing Oak Leaf Powder on Tomato Growth and Biochemical Characteristics under Water Stress Conditions

Nawroz Abdul-razzak Tahir [1,*], Kamaran Salh Rasul [1], Djshwar Dhahir Lateef [2] and Florian M. W. Grundler [3]

1   Horticulture Department, College of Agricultural Engineering Sciences, University of Sulaimani, Sulaimani 46001, Iraq
2   Crop Science and Biotechnology Department, College of Agricultural Engineering Sciences, University of Sulaimani, Sulaimani 46001, Iraq
3   INRES—Molecular Phytomedicine, University of Bonn, Karlrobert-Kreiten-Str. 13, D-53115 Bonn, Germany
*   Correspondence: nawroz.tahir@univsul.edu.iq; Tel.: +964-7701965517

**Abstract:** Drought stress is one of the most significant abiotic stresses on the sustainability of global agriculture. The finding of natural resources is essential for decreasing the need for artificial fertilizers and boosting plant growth and yield under water stress conditions. This study used a factorial experimental design to investigate the effects of oak leaf extract, biofertilizer, and soil containing oak leaf powder on the growth and biochemical parameters of four tomato genotypes under water stress throughout the pre-flowering and pre-fruiting stages of plant development. The experiment had two components. The first component represented the genotypes (two sensitive and two tolerant), while the second component represented the treatment group, which included irrigated plants (SW), untreated and stressed plants (SS), treated plants with oak leaf powder and stressed (SOS), treated plants with oak leaf powder and oak leaf extract and stressed (SOES), and treated plants with oak leaf powder and biofertilizers and stressed (SOBS). When compared with irrigated or control plants, drought stress under the treatments of SS, SOS, SOES, and SOBS conditions at two stages and their combination significantly lowered shoot length (12.95%), total fruit weight per plant (33.97%), relative water content (14.05%), and total chlorophyll content (26.30%). The reduction values for shoot length (17.58%), shoot fresh weight (22.08%), and total fruit weight per plant (42.61%) were significantly larger in two sensitive genotypes compared with tolerant genotypes, which recorded decreasing percentages of 8.36, 8.88, and 25.32% for shoot length, shoot fresh weight, and total fruit weight per plant, respectively. Root fresh weight and root dry weight of genotypes treated with SS, SOS, SOES, and SOBS, on the other hand, increased in comparison with control plants. Tomato fruits from stressed plants treated with SS, SOS, SOES, and SOBS had considerably higher levels of titratable acidity, ascorbic acid, and total phenolic compounds than irrigated plants during all stress stages. Under water stress conditions, the addition of oak leaf powder to soil, oak leaf extract, and biofertilizer improved the biochemical content of leaves in all genotypes. Furthermore, leaf lipid peroxidation was lower in plants treated with SOES and SOBS, and lower in the two tolerant genotypes than in the two susceptible genotypes. In conclusion, the application of SOS, SOES, and SOBS demonstrated a slight decrease in some morpho-physiological and fruit physicochemical traits compared with SS treatment. However, the application of oak leaf powder and oak leaf extract can be described as novel agricultural practices because they are low-cost, easy to use, time-consuming, and can meet the growing demands of the agricultural sector by providing environmentally sustainable techniques for enhancing plant resistance to abiotic stress. The usage of the combination of leaf crude extract, oak leaf powder, and arbuscular mycorrhizal fungus should be investigated further under stress conditions.

**Keywords:** drought; *Solanum lycopersicum*; biostimulation; plant tissue; plant response; enhancement of tolerance

## 1. Introduction

The tomato (*Solanum lycopersicum* L.) belongs to the Solanaceae family, which includes nearly 2800 species, and is one of the world's most important vegetables and crops [1,2]. Its production has increased continuously, reaching nearly 186 million tons of fresh fruit in 2020 [3]. It is consumed as a fresh or processed fruit because of its high nutritional value, which includes vitamins, folate, and phytochemicals [4]. Tomatoes are also considered a perfect fleshy fruit model system because they can be easily grown under different conditions, have a short life cycle, and have simple genetics owing to their small genome and lack of gene duplication [5].

Water resources around the world have decreased as a result of climate change and global warming. Agriculture productivity is significantly impacted by water constraints around the world [6]. The plant's internal water content is affected by low soil water availability, which inhibits its physiological and biochemical functions. Despite the tomato's economic importance, it is susceptible to drought stress, especially during its blooming and fruit enlargement phases [7,8], which prevents seed germination, slows down plant development, and lowers fruit yields [9]. Additionally, little is known about the crucial role of stress-responsive genes, the processes behind their response to abiotic pressures, and the mechanisms underlying their response to biotic challenges [10].

An understanding of how plants respond to fluctuations in environmental conditions is crucial for predicting plant and ecosystem responses to climate change [11]. The plant's response to drought stress is highly dependent on the duration and severity of the stress, but is also influenced by the plant's genotype and its developmental stage [12]. The plants change their cellular activities by producing different defense mechanisms in response to water stress. Drought causes osmotic stress, which can result in turgor loss, membrane deterioration, protein degradation, and often high amounts of reactive oxygen species (ROS), which cause tissue oxidative damage [11]. The antioxidant enzyme systems are produced by some antioxidant enzymes and osmotic substances such as soluble sugars, proteins, and free prolyls, which scavenge these ROS and protect macromolecules in plant cells [13]. Plants adopt different strategies, including the accumulation of some substances with the capability to retain water, such as proline, compatible solutes, and those that evade water deficits by modifying water consumption such as root system traits and C3/C4 or CAM photosynthesis [11,14]. Stressed plants produce some important metabolites, like organic acids, polyamines, amino acids, and lipids, which moderately alleviate stress by acting as osmoregulators, antioxidants, and defense compounds [15]. Some protein kinases are turned on in most plants when they are under water stress. These include mitogen-activated protein kinases (MAPKs), calcium-dependent protein kinases (CDPKs), calcineurin B-like (CBL)-interacting protein kinases (CIPKs), and members of the sucrose non-fermenting-1 (SNF1)-related protein kinase 2 (SnRK2) family [16,17].

The addition of plant tissue to soil improves soil quality by reducing the risk of soil erosion and increasing crop yields [18]. Plant tissue application also plays a crucial role in sustaining and improving the chemical, physical, and biological properties of the soil by providing mineral nutrients and protecting the soil's water content [19] and may have an effect on plant water uptake [20]. Silicon (Si) is a nutritional mineral in the plant residue that promotes plant growth and development, particularly under dry conditions. Si ameliorates osmotic and ionic stressors associated with drought [21]. Si-treated plants maintained stomatal conductance and transpiration rate, leaf relative water content, as well as root and whole-plant hydraulic conductivity [22].

Natural biofertilizer is a product made from living microorganisms that are extracted from cultivated or root soil. It is safe for the environment and soil health, and it is essential for atmospheric nitrogen fixation and phosphorus solubilization, which leads to increased nutrient uptake and tolerance to drought and moisture stress [23]. Rhizobacteria that promote plant growth (PGPR) are a favorable interaction between microbes and plants that can speed up plant growth. One category of rhizobacteria consists of *Bacillus* species, which support plant growth, increase nutrient availability, increase the production of

plant hormones, generate volatiles, and lessen the effects of drought [24,25]. Several studies analyzed the chemical profile of leaves of different oak species and confirmed the presence of several chemical elements, including phenolic, flavonoid, and terpenoid substances. In addition, they demonstrated significant radical scavenging, antibacterial, and antitopoisomerase activity [26–30]. To the best of our knowledge, no research has been conducted on the use of oak leaf extract and powder as biostimulator factors in water stress situations.

Owing to the presence of high amounts of chemical compounds related to growth and antioxidant activity, the hypothesis of this study was to test and determine the biological activity of oak tissues. The goal of this study was to determine the effects of oak leaf extract, biofertilizer, and soil incorporating oak leaf powder on the growth and biochemical traits of four tomato genotypes under water stress conditions during two stages of plant development. This research will help farmers find new ways to use oak leaf powder and extract because they are cheap, easy to use, and do not take much time. They can also meet the growing needs of the agricultural industry by providing environmentally friendly ways to make plants more resistant to abiotic stress.

## 2. Materials and Methods

### 2.1. Plant Materials

This study used two susceptible tomato genotypes, Braw and Yadgar, and two tolerant tomato genotypes, Raza Pashayi and Sandra, based on the results of in vitro tests of 64 tomato genotypes to drought stress by polyethylene glycol-MW 6000 (unpublished data). The tomato genotypes were collected from the Agricultural Research Center of the Ministry of Agriculture and Water Resources in Kurdistan, Iraq.

### 2.2. Experimental Design Components, Plant Treatement, and Growth Conditions

The experiment is divided into three groups. The plants in Group 1 were stressed before flowering. The plants in the second group were stressed prior to fruiting. The third category includes plants that were stressed before flowering and fruiting. To conduct this investigation, a factorial completely randomized design (CRD) with two components was applied. The first component represented tomato genotypes (two sensitive and two tolerant) and the second component represented the treatment group, which consisted of irrigated plants (SW), stressed plants (SS), stressed plants + oak leaf powder (SOS), stressed plants + oak leaf powder + oak leaf extract (SOES), and stressed plants + oak leaf powder + biofertilizers (SOBS). Seeds of four genotypes were planted in plastic trays in a plastic house. Fully developed and healthy oak leaves (*Quercus aegilops* Oliv.) were gathered at the vegetative stage on 17 May 2021, dried, and ground into powder for the SOS, SOES, and SOBS treatments. The seedlings were transplanted into the plastic pots (40 cm in height and 18 cm in diameter). The pots for SW and SS treatments contained only 10 kg of soil, whereas the pots for SOS, SOES, and SOBS contained 10 kg of soil and 80 g of oak leaf powder. Each treatment was composed of eight replications (eight plants) (Figure S1).

To make the extract of oak leaf, 60 g of powdered oak leaves were dissolved in 1 L of distilled water, shaken for 3 h, and then incubated overnight at 5 °C [31,32]. After centrifuging for 30 min at 4000 rpm, the supernatant was collected and diluted (1:29 *v/v*) with distilled water. This extract was applied four times by foliar spray before flowering (first stress stage) and fruiting (second stress stage) with three-day intervals. Leaf extract was sprayed before flowering on 7 June, 10 June, 13 June, and 16 June 2021 and before fruiting on 15 July, 18 July, 21 July, and 24 July 2021. For biofertilizer treatment, 40 mg per plant of Fulzyme Plus (JH Biotech.; Inc.; USA) was applied as fertigation three times in 15 days. This biofertilizer consisted of beneficial bacteria like *Bacillus subtilis* and *Pesudomonas putida* ($2 \times 10^{10}$ g); enzymes like protease, amylase, lipase, and chitinase; and hormones like gibberellin (0.3%) and cytokinin (0.3%). Water stress at 40% of field capacity was applied before flowering (the first stress stage) for six days and fruiting (the second stress stage) for four days [7]. The plants grew over the spring and summer sessions of 2021. The



average daytime and nighttime relative humidity in the greenhouse during the experiment was 42.84/17.17% and the average temperature was 39.55/23.59 °C. Plants were kept in a regular photoperiod with 14 h of natural light per day. Weeds were physically eliminated during the plant's growing stage, and unhealthy or dried leaves were taken out.

The soil in the experiment was silty clay in texture, with an EC of 0.61 dS m$^{-1}$, a pH of 7.5, an organic matter content of 17.79 g kg$^{-1}$, a total nitrogen content of 15.56 g kg$^{-1}$, a phosphorus content of 4.44 mg kg$^{-1}$, an available potassium content of 0.16 meq L$^{-1}$, and an exchangeable phosphorus content of 0.2 mg kg$^{-1}$.

### 2.3. Evaluation of Morphological and Physiological Parameters

Plant morphological data from eight plants per treatment, including shoot length (SL in cm), shoot fresh weight (SFW in g), shoot dry weight (SDW in g), root length (RL in cm), root fresh weight (RFW in g), root dry weight (RDW in g), and fruit weight per plant (FWT in g), were measured at the end of the stress period. The total chlorophyll content of the leaves of eight plants (TCC in SPAD) was determined using a SPAD-meter at the end of the stress period. Using the method outlined by Lateef et al. [33], the relative water content (RWC in %) of the leaves was estimated using six leaves from eight tomato plants harvested at the end of the stress period.

### 2.4. Tomato Leaves' and Fruits' Collection

At the end of the stress point, fresh tomato leaves were collected, ground using liquid nitrogen, and frozen at −20 °C for use in biochemical investigations. Tomato fruits were hand-harvested at full maturity and stored at −20 °C for use in tomato fruit quality tests.

### 2.5. Moisture Content, Titratable Acidity, and Total Soluble Solid Measurement

The moisture content (MC) of eight plants was estimated by weighing 10 g of fresh tomato fruit and then drying the samples at 70 °C for 72 h until a consistent weight was achieved. The weight of the dry samples was determined and the MC percentage was calculated using the following equation [34,35]:

$$MC \ (\%) = \frac{FW - DW}{DW} \times 100$$

where MC is the moisture content of tomato fruit, FW is the fresh weight of tomato fruit, and DW is the dry weight of tomato fruit.

Titratable acidity (TA) was determined by combining 3 mL of tomato juice with two to three drops of phenolphthalein and titrating the mixture with 0.1 N NaOH [36]. TA was computed using the following formula:

$$TA \ (\%) = \frac{\text{Volume of titrant} \ \times \ N \ (NaOH) \times \text{Acid equivilent}}{\text{Volume of used juice} \times 1000} \times 100$$

Total soluble solids (TSSs, Brix) was determined using a digital refractometer [34,35]. Fruits of six plants from each level of treatments were subjected to this test.

### 2.6. Measurement of Biochemical Traits
#### 2.6.1. Ascorbic Acid Content (ASC)

Ascorbic acid content (ASC) was determined by combining 0.4 g of powdered tomato fruit tissue with 1300 μL of 1% (*w/v*) HCl and vigorously shaking the mixture for 30 min. The mixture was centrifuged for 10 min at 13,000× *g* rpm and the supernatant was collected. The supernatant was mixed with 1900 μL of 1% (*v/v*) HCl and measured at 243 nm against a blank containing 1% (*v/v*) of HCl [37].

### 2.6.2. Carotenoid Content (CAC)

One gram of powdered tomato fruit tissue was mixed with 1000 µL of 100% methanol, and the mixture was incubated overnight at 5 °C. After centrifuging the samples for 8 min at 13,000× $g$ rpm, 500 µL of the supernatant was collected and mixed with 1500 µL of 100% methanol. At 470 nm, the sample was read against a blank of 100% methanol [38].

### 2.6.3. Soluble Sugar Content (SSC)

Using the method described by Lateef et al. [33], the concentration of soluble sugar in fresh leaves and fruits was determined.

### 2.6.4. Proline Content (PC)

The proline content of the fresh leaves was determined using the method of Lateef et al. [33].

### 2.6.5. Total Phenolic Content (TPC)

According to Lateef et al. [33], fresh fruits and leaves were tested for their total phenolic content (TPC).

### 2.6.6. Antioxidant Compound Capacity (AC)

The antioxidant capacity was evaluated by combining 0.1 g of ground fresh leaves with 1 mL of 60% (*v/v*) acidic methanol (%99 methanol + %1 HCl). After shaking the mixture for 10 min, the sample was incubated at 5 °C overnight. The mixture was centrifuged for 15 min at 12,000× $g$ rpm to collect the supernatant. Using the 1-diphenyl-2-picrylhydrazyl (DPPH) method as described by Lateef et al. [33], the antioxidant capacity of supernatant (extract) was assessed.

### 2.6.7. Antioxidant Enzyme Activity

The activities of guaiacol peroxidase (GPA) and catalase (CAT) were determined using the procedures reported by Lateef et al. [33].

### 2.6.8. Lipid Peroxidation Assays

As a biomarker of membrane oxidative damage caused by the water stress, the concentration of malondialdehyde (MDA), which is the final product of lipid peroxidation, was measured [39]. This experiment was initiated by mixing an amount of grinded powder leaves (0.4 g) with 2 mL of Tris-HCl buffer solution (pH 7.4) comprising 1.5% (*w/v*) of polyvinylpyrrolidone (PVP). Then, the mixture was shaken well for a duration of 10 min. Afterwards, the solution mixture was centrifuged at 10,000× $g$ rpm for half an hour. All of the upper layers were then taken and transferred to a glass tube. Following that, 2 mL of 0.5% (*w/v*) thiobarbituric acid in 20% trichloroacetic acid (*w/v*) was mixed with the supernatant and boiled for 31 min at 95 °C in a water bath. After heating, the samples were immediately placed in a cold-water bath to stop the reactions, and the pinkish color appeared among the samples. The reaction mixture, after centrifugation at 4000× $g$ rpm for 12 min, was measured at two different wavelengths, 532 and 600 nm. The first measurement is a true measurement of the sample, while the second is for correcting unclear turbidity by subtracting the value of absorbance at 600 nm. The concentration of lipid peroxidation (LP) was stated in nmol g$^{-1}$ seedling fresh weight:

$$LP = \frac{AB532 - AB600 \times 1000 \times VL}{EC \times WE}$$

where AB532 is the absorbance at 532 nm, AB600 is the absorbance at 600 nm, VL is the volume of extract (mL), WE is the fresh weight of the sample (g), and EC is the extinction coefficient of 155 mM$^{-1}$cm$^{-1}$.

### 2.7. GC-MS Analysis of Oak Leaf Extract

The chemical components of oak leaf extract were identified using an Agilent 7890 B gas chromatograph and an Agilent 5977 mass spectrometer, both manufactured by MSD, USA. HP-5MS UI capillary column (30 m × 0.25 × 0.25 mm) fused with 5% phenyl methyl siloxane and a splitless injector were used in a gas chromatograph. The initial temperature in the column oven was 40 °C, held steady for 60 s, and then increased to 300 °C at a rate of 10 °C per minute. To do this, we used a constant flow rate of 1 mL/min of helium as the carrier gas and heated the injector to 290 °C. In the splitless model, the injection volume was 1 mL, the purge flow was 3 mL/min, the total flow was 19 mL/min, and the pressure was 7.0699 psi. The mass spectrometer was run with the help of the Mass Hunter GC/MS Acquisition software and the Mass Hunter qualitative program, which scanned fragments in the range of 35 m/z to 650 m/z. The interface temperature (MSD transfer line) was set at 290 °C, the ionization source temperature was set at 230 °C, and the quad temperature was set at 150 °C. The solvent cut time began at 4 min and ended between 35 and 40 min.

### 2.8. Statistical Data Analysis

XLSTAT version 2019.2.2 (Boston, USA) was used to run statistical analyses (two-way analysis of variance, Duncan's multiple range test, and principal component analysis (PCA)) for assessing the data obtained in this study at $p \leq 0.05$ [40]. The trait index was calculated by the following formula [41]:

$$\text{Trait index (\%)} = \frac{(\text{Mean of treated and stressed plants} - \text{Mean of irrigated plants})}{\text{Mean of irrigated plants}} \times 100$$

The values of all studied traits are represented by the mean ± standard deviation (SD). Each value is the average of three replications for physicochemical parameters and eight replications for morpho-physiological traits.

## 3. Results

### 3.1. Effect of Various Treatments on the Morpho-Physiological and Fruit Physicochemical Traits of Tomato under Water Stress

Plant development and growth are essentially the results of cell division, cell enlargement, and differentiation, and they are regulated by a variety of genetic, physiological, ecological, and morphological processes, as well as their interconnections [42]. The analysis of variance on morphological characters, relative water content (RWC), and total chlorophyll content (TCC) in the first stress stage (before flowering), the second stress stage (before fruiting), and their combinations revealed that treatments had a significant effect (Table S1 and Figures S2 and S3). When compared with control plants, all levels of treatment resulted in a significant percentage decrease in shoot length (SL), shoot fresh weight (SFW), shoot dry weight (SDW), fruit weight per plant (FWT), relative water content (RWC), and total chlorophyl content (TCC). In comparison with control plants, the stressed plant group (SS) that was not exposed to powdered oak tissue, oak leaf extract, or biofertilizer at any stage had the highest decline percentages for all traits (Table 1).

According to the results of the interaction, Braw under SOBS application resulted in the highest increasing percentages of SFW (33.35%), SDW (51.30%), and RFW (145.06%) compared with the irrigated plants (SW) during the first stress stages, while Yadgar under untreated and stressful conditions (SS) resulted in the maximum decreasing values for FWT (50.38%) and RWC (18.72%) (Table S4). The interaction results showed that, during the second stress stages, Braw under SOBS application contributed to the greatest increases in SFW (5.03%), SDW (29.64%), and RFW (258.68%) compared with the control conditions, while Yadgar (48.30%) and Sandra (48.11%) under the SOS condition caused the greatest decreases in FWT and TCC. As per Table S4, the interaction outcomes demonstrated that the Sandra genotype under SOBS application contributed to the highest increases in SDW (2.74%), and RDW (255.70%) compared with SW conditions, and that Yadgar under the

SS condition caused the greatest decreases in SL (26.52%) and FWT (63.89%) during the combination of both stress stages.

**Table 1.** Effect of oak leaf powder, oak leaf extract, and biofertilizer on the morpho-physiological characteristics of tomato plants at various stress stages. Positive and negative values signify increasing and declining, respectively.

| | Increasing and Decreasing Percentages Compared with Irrigated Plants in the First Stress Stage | | | | | | | | |
|---|---|---|---|---|---|---|---|---|---|
| Treatment | SL (%) | SFW (%) | SDW (%) | RL (%) | RFW (%) | RDW (%) | FWT (%) | RWC (%) | TCC (%) |
| SOBS | −6.70 a ± 5.15 | 1.90 a ± 20.80 | 7.72 a ± 27.31 | 4.94 a ± 16.58 | 74.93 a ± 50.94 | 99.21 a ± 84.92 | −27.30 ab ± 9.53 | −11.94 ab ± 3.26 | −24.30 b ± 12.18 |
| SOES | −6.54 a ± 8.47 | −5.78 b ± 7.59 | −3.24 b ± 8.14 | 0.03 ab ± 12.43 | 44.00 b ± 58.76 | 43.76 ab ± 117.82 | −21.36 a ± 16.97 | −9.44 a ± 8.07 | −9.80 a ± 18.67 |
| SOS | −7.13 a ± 5.52 | −8.93 b ± 6.53 | −11.53 c ± 6.95 | −4.57 b ± 15.55 | 30.05 b ± 26.34 | 22.51 b ± 20.64 | −28.59 b ± 10.07 | −14.24 bc ± 3.89 | −33.34 b ± 8.43 |
| SS | −13.52 b ± 6.56 | −24.76 c ± 15.11 | −22.98 d ± 14.67 | −16.67 c ± 11.37 | 16.16 b ± 29.80 | 26.11 b ± 52.72 | −32.58 b ± 11.58 | −16.75 c ± 4.69 | −31.30 b ± 10.72 |
| | Increasing and Decreasing Percentages Compared with Irrigated Plants in the Second Stress Stage | | | | | | | | |
| Treatment | SL (%) | SFW (%) | SDW (%) | RL (%) | RFW (%) | RDW (%) | FWT (%) | RWC (%) | TCC (%) |
| SOBS | −9.28 a ± 4.57 | −7.95 a ± 13.31 | 3.02 a ± 18.52 | 15.70 a ± 17.39 | 107.57 a ± 104.78 | 121.80 a ± 91.08 | −28.49 a ± 13.46 | −10.12 a ± 4.59 | −24.18 b ± 12.07 |
| SOES | −9.04 a ± 5.33 | −14.16 b ± 10.09 | −5.18 b ± 11.10 | 5.96 b ± 15.92 | 94.83 a ± 86.72 | 104.54 a ± 82.27 | −30.57 a ± 11.03 | −8.87 a ± 7.42 | −16.85 a ± 12.49 |
| SOS | −13.00 a ± 6.63 | −20.21 c ± 10.32 | −14.03 c ± 8.70 | −11.47 c ± 14.93 | 30.54 b ± 39.65 | 36.60 b ± 26.44 | −35.13 b ± 10.26 | −10.88 a ± 3.99 | −33.29 c ± 10.34 |
| SS | −20.56 b ± 8.62 | −29.05 d ± 15.83 | −25.14 d ± 14.76 | −12.84 c ± 11.43 | 29.59 b ± 39.99 | 31.48 b ± 67.26 | −37.64 b ± 11.47 | −18.14 b ± 6.84 | −31.55 c ± 10.75 |
| | Increasing and Decreasing Percentages Compared with Irrigated Plants in the First and Second Stress Stages | | | | | | | | |
| Treatment | SL (%) | SFW (%) | SDW (%) | RL (%) | RFW (%) | RDW (%) | FWT (%) | RWC (%) | TCC (%) |
| SOBS | −13.95 a ± 7.95 | −15.22 b ± 12.10 | −9.28 a ± 11.43 | 8.52 a ± 17.42 | 92.02 a ± 83.34 | 101.46 a ± 100.45 | −41.10 a ± 13.87 | −15.59 a ± 6.65 | −26.22 b ± 11.54 |
| SOES | −14.57 a ± 7.57 | −9.14 a ± 12.32 | −8.57 a ± 8.88 | 2.52 ab ± 14.18 | 89.63 a ± 76.02 | 100.71 a ± 106.78 | −39.72 a ± 12.98 | −13.70 a ± 6.65 | −16.24 a ± 14.80 |
| SOS | −17.50 a ± 8.64 | −16.48 b ± 13.13 | −13.05 b ± 10.59 | −3.73 b ± 9.00 | 51.78 b ± 52.23 | 62.56 b ± 117.59 | −40.04 a ± 11.37 | −16.87 a ± 6.45 | −32.91 c ± 8.37 |
| SS | −23.80 b ± 9.89 | −36.00 c ± 23.39 | −27.49 c ± 16.15 | −12.98 c ± 6.97 | 27.08 b ± 35.52 | 37.11 c ± 86.64 | −45.10 b ± 13.80 | −22.06 b ± 5.42 | −35.64 c ± 10.40 |

SL: shoot length, SFW: shoot fresh weight, SDW: shoot dry weight, RL: root length, RFW: root fresh weight, RDW: root dry weight, FWT: fruits weight per plant, RWC: relative water content, TCC: total chlorophyl content, SS: stressed plants that had not been treated, SOS: stressed plants that had been treated with oak leaf powder, SOES: stressed plants that had been treated with oak leaf powder and oak leaf extract, SOBS: stressed plants that had been treated with oak leaf powder and biofertilizers. Duncan's multiple range test at $p \leq 0.05$ indicates that any mean values sharing the same letter in the same column are not statistically significant. The value is represented by trait index ± standard deviation (SD). Each value is the average of eight measurements.

The analysis of variance (ANOVA) of the data reported a significant influence of the treatment on the fruit's physicochemical properties (Table S2). As stated in Table 2, the titratable acidity (TA). ascorbic acid content (ASC), and total pheolic content (TPC) responded positively to different levels of treatment in all stages of growth. In the first stress stage, the highest increasing percentages of TA, ASC, and TPC were obtained by the treatments SS (11.23%), SOBS (23.50%), and SOES (11.10%), respectively. The TA, ASC, and TPC responded favorably to various treatments during the second stress stage. The highest increasing TA (12.63%), ASC (18.49%), and TPC (12.21%) values were seen in the treatments SS, SOES, and SOBS, respectively. Similarly, when two stress measures were combined, the same results were found. In the SS and SOES applications, the highest percentage increases in TA (19.05%), ASC (13.11%), and TPC (10.42%) were shown. Under all stress conditions, a decreasing amount was also observed in the moisture content (MC), total soluble solids (TSSs), and carotenoid content (CAC). The SS application showed the largest decline in percentage in MC, TSS, and CAC. With the first stress stage, the soluble sugar content (SSC) decreased by 3.27 and 2.78% under SOBS and SOES conditions, respectively. The SSC responded favorably to the SOBS and SOES applications during the second stress stage,

increasing by 1.68 and 2.73%, respectively. Under all levels of treatment (SS, SOS, SOES, and SOBS), the SSC values for both stress stages together decreased.

**Table 2.** Influence of oak leaf powder, oak leaf extract, and biofertilizer on the fruit physicochemical parameters of tomato plants at different stress stages. Increasing and decreasing are labeled by a positive and negative value, respectively.

| Increasing and Decreasing Percentages Compared with Irrigated Plants in the First Stress Stage | | | | | | | |
|---|---|---|---|---|---|---|---|
| Treatment | MC | TA | TSS | ASC | CAC | SSC | TPC |
| SOBS | −0.67 a ± 0.45 | 2.64 bc ± 6.43 | −1.73 a ± 4.35 | 23.50 a ± 12.40 | −3.02 b ± 4.43 | 3.27 a ± 7.73 | 8.55 b ± 12.38 |
| SOES | −0.59 a ± 0.35 | 1.03 c ± 6.81 | −2.08 a ± 3.60 | 22.10 b ± 14.06 | −1.80 a ± 4.25 | 2.78 a ± 8.68 | 11.10 a ± 10.93 |
| SOS | −0.87 b ± 045 | 5.82 b ± 5.12 | −4.43 b ± 4.29 | 15.92 c ± 11.81 | −5.35 c ± 6.19 | −2.34 b ± 7.72 | 9.06 b ± 10.41 |
| SS | −1.28 c ± 0.83 | 11.23 a ± 6.53 | −6.67 c ± 4.37 | 6.41 d ± 10.22 | −7.64 d ± 7.61 | −8.28 c ± 11.84 | 5.06 c ± 10.63 |
| Increasing and Decreasing Percentages Compared with Irrigated Plants in the Second Stress Stage | | | | | | | |
| Treatment | MC | TA | TSS | ASC | CAC | SSC | TPC |
| SOBS | −0.65 a ± 0.046 | 3.45 b ± 7.54 | −1.59 a ± 3.23 | 17.22 b ± 18.89 | −0.03 a ± 6.67 | 1.68 b ± 10.98 | 12.21 a ± 14.70 |
| SOES | −0.55 a ± 0.039 | 2.12 b ± 7.50 | −2.42 a ± 3.26 | 18.49 a ± 16.69 | −2.40 b ± 7.74 | 2.73 a ± 9.89 | 11.37 a ± 14.15 |
| SOS | −0.87 b ± 0.44 | 6.00 b ± 5.91 | −4.21 b ± 4.64 | 13.40 c ± 17.44 | −4.70 c ± 8.46 | −2.71 c ± 9.85 | 9.37 b ± 12.41 |
| SS | −1.23 c ± 0.78 | 12.63 a ± 11.80 | −6.51 c ± 5.14 | 2.29 d ± 12.86 | −7.80 d ± 10.15 | −8.35 d ± 13.56 | 4.68 c ± 9.02 |
| Increasing and Decreasing Percentages Compared with Irrigated Plants in the First and Second Stress Stages | | | | | | | |
| Treatment | MC | TA | TSS | ASC | CAC | SSC | TPC |
| SOBS | −0.90 a ± 0.58 | 6.65 c ± 6.14 | −3.86 a ± 4.53 | 12.73 a ± 16.38 | −2.58 a ± 9.90 | −1.49 a ± 10.12 | 9.12 b ± 13.75 |
| SOES | −0.86 a ± 0.59 | 7.19 c ± 5.68 | −4.27 a ± 4.50 | 13.11 a ± 17.24 | −5.12 b ± 9.91 | −1.69 a ± 9.79 | 10.42 a ± 14.22 |
| SOS | −1.21 b ± 0.74 | 11.47 b ± 5.94 | −5.88 b ± 5.32 | 7.04 b ± 17.51 | −6.92 c ± 10.56 | −6.41 b ± 10.34 | 7.65 c ± 12.05 |
| SS | −1.55 c ± 1.01 | 19.05 a ± 11.13 | −8.54 c ± 5.74 | −3.73 c ± 14.17 | −10.10 d ± 11.68 | −12.31 c ± 13.16 | 1.78 d ± 10.46 |

MC: moisture content, TA: titratable acidity, TSS: total soluble solids, ASC: ascorbic acid content, CAC: carotenoid content, SSC: soluble sugar content, TPC: total phenolics content, SS: stressed plants that had not been treated, SOS: stressed plants that had been treated with oak leaf powder, SOES: stressed plants that had been treated with oak leaf powder and oak leaf extract, SOBS: stressed plants that had been treated with oak leaf powder and biofertilizers. Duncan's multiple range test at $p \leq 0.05$ indicates that any mean values sharing the same letter in the same column are not statistically significant. The value is represented by trait index ± standard deviation (SD). Each value is the average of three measurements.

A multivariate analytic technique called principal component analysis (PCA) is used to evaluate the similarity between the levels of treatment. Additionally, it is also used to determine the relationship between attributes. In total, 16 determined variables concerning the morpho-physiological and fruit physicochemical traits under four levels of treatment were subjected to a principal component analysis. Based on an eigenvalue > 1, we extracted a total of two first components with a cumulative distribution of 95.63% (85.05% for the first component and 11.59% for the second component), 96.53% (90.26% for the first component and 6.27% for the second component), and 97.04% (92.95% for the first component and 4.09% for the second component) for the first, second, and their combination stress stages, respectively (Figure 1). Different distributions of studied traits and treatments were observed on the PCA plot. Under first stress stage, the most notable contributors to the observed variance along PC1 were SL, SFW, RL, RWC, MC, TA, TSS, ASC, CAC, and SSC. However, the greatest amount of variance along PC2 was caused by SDW, RFW, RDW, FWT, TCC, and TPC (Figure 1A). The most noteworthy contributions to the observed variance along PC1 during the second stress stage were SL, SFW, SDW, FWT, MC, TSS, CAC, SSC, and TPC. Nevertheless, RL, RFW, RDW, RWC, TCC, MC, TA, and ASC were responsible for the bulk of the variation along PC2 (Figure 1B). Under both stress stages, the SL, SDW, RFW, RDW, RWC, TCC, MC, TA, TSS, ASC, SSC, and TPC were the major contributors to the observed variance along PC1. SFW, RL, FWT, and CAC, on the other hand, were responsible for the majority of the variation along PC2 (Figure 1C).

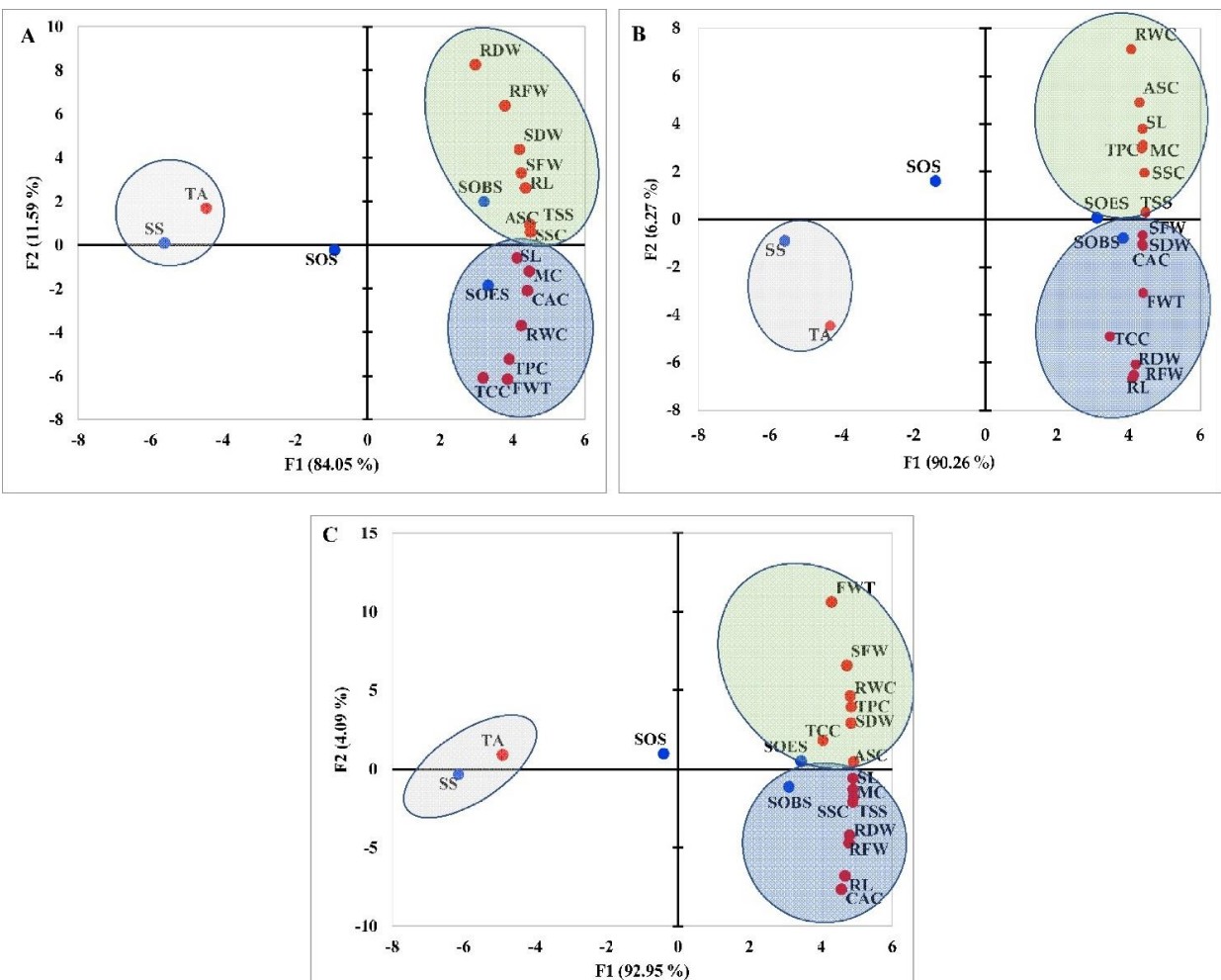

**Figure 1.** PCA plot showing the distribution of various morpho-physiological and fruit physico-chemical traits and treatments under first (**A**), second (**B**), and both (**C**) stress conditions. SL: shoot length, SFW: shoot fresh weight, SDW: shoot dry weight, RL: root length, RFW: root fresh weight, RDW: root dry weight, FWT: fruits weight per plant, RWC: relative water content, TCC: total chlorophyll content, MC: moisture content, TA: titratable acidity, TSS: total soluble solids, ASC: ascorbic acid content, CAC: carotenoid content, SSC: soluble sugar content, TPC: total phenolic content, SS: stressed plants that had not been treated, SOS: stressed plants that had been treated with oak leaf powder, SOES: stressed plants that had been treated with oak leaf powder and oak leaf extract, SOBS: stressed plants that had been treated with oak leaf powder and biofertilizers. F1 and F2 represent the first and second components, respectively.

The application of powdered oak leaf, leaf oak extract, and biofertilizers reduced titratable acidity (TA) in fruit in all stress stages compared with the untreated plant under stress conditions and formed the first group in the left of the PCA plot (brown outline). During the first stress stage, the characteristics of the plants treated with SOBS with high percentage values of RL, SFW, SDW, RFW, RDW, TSS, ASC, and SSC were included in the second group on the upper right quadrant (green outline) of the PCA plot. The third group in the lower right quadrant (blue outline) of the PCA plot is made up of attributes in SOES-treated plants with high SL, RWC, TCC, FWT, MC, CAC, and TPC values. Under the second stress stage, the characteristics of the plants treated with SOES that had high values of SL, TSS, MC, SSC, TPC, RWC, and ASC formed the second group in the upper right quadrant (green circle) of the PCA plot. Furthermore, the traits in the plants treated with SOBS with high percentage values of RL, RFW, RDW, SFW, SDW, FWT, TCC, and CAC were included in the third group on the lower right quadrant (blue circle) of the PCA

plot. In the combination of both stress stages, traits with high percentage values of SFW, SDW, FWT, RWC, TCC, TPC, and ASC in plants treated with SOES comprised the second group in the upper right quadrant (green circle) of the PCA plot. The third group was in the lower right quadrant (blue outline) of the PCA plot. It was made up of plants treated with SOBS and having high values of SL, RL, RFW, RDW, MC, CAC, TSS, and SSC.

### 3.2. Influence of Genotypes on the Morpho-Physiological and Physicochemical Characteristics of Tomato Fruit under Application of SS, SOS, SOES, and SOBS

Under conditions of water stress, analysis of variance (ANOVA) revealed highly significant genotype effects on the morpho-physiological traits of the first stress stage (before blooming), the second stress stage (before fruiting), and their combinations (Table S1). Shoot length (SL), shoot fresh weight (SFW), shoot dry weight (SDW), fruit weight per plant (FWT), relative water content (RWC), and total chlorophyll content (TCC) were all significantly lower in all genotypes as compared with control plants. SL (13.13%), SFW (17.96%), SDW (17.83%), FWT (42.63%), and RWC (15.68%) exhibited the largest decreasing percentages in the stressed Yadgar genotype. In all stress stages, the tolerant genotypes (Raza Pashayi and Sandra) had lower decreasing amounts of SL and FWT than the sensitive genotypes (Braw and Yadgar) (Table 3). Root fresh weight (RFW) and root dry weight (RDW) demonstrated high increasing percentages in four genotypes for all stress levels under water stress circumstances.

**Table 3.** Impact of tomato genotypes treated with oak leaf powder, oak leaf extract, and biofertilizer at different stress stages on the morpho-physiological traits. Increasing and declining percentages are represented by positive and negative values, respectively.

| Genotypes | SL (%) | SFW (%) | SDW (%) | RL (%) | RFW (%) | RDW (%) | FWT (%) | RWC (%) | TCC (%) |
|---|---|---|---|---|---|---|---|---|---|
| **Increasing and Decreasing Percentages Compared with Irrigated Plants during the First Stress Stage** | | | | | | | | | |
| Raza Pashayi | −4.98 a ± 5.24 | −5.75 a ± 2.20 | −3.79 b ± 1.95 | −6.52 bc ± 11.01 | 34.19 b ± 17.88 | 53.76 ab ± 16.65 | −19.47 a ± 6.44 | −11.75 ab ± 4.63 | −31.53 b ± 8.38 |
| Sandra | −5.14 a ± 5.81 | −8.03 a ± 7.99 | −9.21 c ± 9.88 | −5.18 b ± 7.93 | 36.34 b ± 60.49 | 98.55 a ± 146. 60 | −21.26 ab ± 14.64 | −10.13 a ± 8.03 | −32.62 b ± 23.06 |
| Braw | −10.65 b ± 6.62 | −5.83 a ± 30.20 | 0.79 a ± 35.77 | −14.79 c ± 13.11 | 66.29 a ± 64.37 | 24.59 b ± 41.84 | −26.48 b ± 5.15 | −14.81 b ± 5.28 | −17.60 a ± 8.38 |
| Yadgar | −13.12 b ± 7.06 | −17.96 b ± 9.03 | −17.83 d ± 9.88 | 10.23 a ± 19.20 | 28.30 b ± 25.01 | 14.68 b ± 16.65 | −42.63 c ± 6.24 | −15.68 b ± 3.01 | −16.98 a ± 11.40 |
| **Increasing and Decreasing Percentages Compared with Irrigated Plants during the Second Stress Stage** | | | | | | | | | |
| Raza Pashayi | −8.71 a ± 4.44 | −6.32 a ± 1.83 | −4.32 a ± 1.17 | −2.97 bc ± 17.02 | 43.51 b ± 27.94 | 76.21 b ± 24.00 | −18.68 a ± 8.55 | −12.57 ab ± 4.76 | −31.19 c ± 7.80 |
| Sandra | −9.35 a ± 5.40 | −13.32 b ± 8.57 | −7.98 a ± 10.95 | −11.49 c ± 10.65 | 53.32 b ± 51.17 | 158.73 a ± 94.25 | −29.87 b ± 4.25 | −8.93 a ± 8.25 | −39.90 d ± 8.22 |
| Braw | −18.44 b ± 9.70 | −21.72 c ± 20.98 | −6.05 a ± 28.75 | −0.54 b ± 20.29 | 150.13 a ± 110.13 | 52.48 b ± 66.73 | −36.66 c ± 3.49 | −14.55 b ± 4.13 | −14.13 a ± 9.99 |
| Yadgar | −15.39 b ± 6.74 | −30.00 d ± 6.22 | −22.97 b ± 7.07 | 12.36 a ± 20.13 | 15.57 b ± 22.11 | 6.99 c ± 11.82 | −46.60 d ± 6.52 | −11.96 ab ± 8.40 | −20.65 b ± 7.58 |
| **Increasing and Decreasing Percentages Compared with Irrigated Plants during the First and Second Stress Stages** | | | | | | | | | |
| Raza Pashayi | −9.97 a ± 3.90 | −6.23 a ± 1.68 | −3.28 a ± 1.61 | 0.14 b ± 11.89 | 36.62 c ± 23.30 | 53.07 b ± 26.86 | −25.90 a ± 5.29 | −22.04 c ± 2.75 | −33.06 c ± 7.58 |
| Sandra | −12.01 a ± 6.33 | −13.64 b ± 9.47 | −10.01 b ± 10.39 | −9.82 c ± 8.32 | 73.29 b ± 45.59 | 238.51 a ± 47.09 | −36.77 b ± 3.01 | −15.54 ab ± 6.41 | −40.04 d ± 6.58 |
| Braw | −27.55 c ± 7.28 | −24.46 c ± 29.40 | −18.78 c ± 18.75 | −4.75 bc ± 14.41 | 143.07 a ± 75.82 | 12.30 c ± 48.49 | −44.83 c ± 4.28 | −18.42 bc ± 5.54 | −15.80 a ± 10.58 |
| Yadgar | −20.29 b ± 6.15 | −32.50 d ± 8.00 | −26.31 d ± 6.58 | 8.77 a ± 16.95 | 7.52 d ± 18.88 | −2.04 c ± 17.34 | −58.46 d ± 5.43 | −12.23 a ± 8.06 | −22.11 b ± 13.26 |

SL: shoot length, SFW: shoot fresh weight, SDW: shoot dry weight, RL: root length, RFW: root fresh weight, RDW: root dry weight, FWT: fruits weight per plant, RWC: relative water content, TCC: total chlorophyl content. Any mean values sharing the same letter in the same column are not statistically significant, according to Duncan's multiple range test at $p \leq 0.05$. The values are represented by the standard deviation of the trait index. Each value is the average of eight measurements. The value is represented by trait index ± standard deviation (SD).

The analysis of variance (ANOVA) of the data obtained for the fruit physicochemical traits found a significant genotype effect (Table S2). According to Table 4, all stress stages contributed to a reduction in the four genotypes' moisture content (MC). The fruit of tolerant genotypes showed higher increasing values in the ASC, CAC, and TPC characteristics than sensitive genotypes under all stress stages. Under all stages of stress, the TA was higher in sensitive genotypes than in tolerant genotypes. Additionally, the Sandra genotype showed an increase in CAC of 2.22, 5.01, and 6.95% for the first, second, and both of them together, respectively.

In accordance with Table S5, the mean pairwise comparison for the interaction of genotypes and different treatments showed that Sandra had the highest increasing percentages in CAC (3.18%) and SSC (16.32%) in the presence of the SOES application, followed by Raza Pashayi with the highest increasing percentages in ASC (36.58%) and TPC (22.43%). With the exception of TA with the treatment of SS and SOS, the Braw genotype reported the highest declining values in all physicochemical parameters under the first stress stage. The Sandra genotype registered the largest percentage increases in TSS (2.63%), ASC (38.21%), CAC (6.02%), and SSC (16.67%) compared with irrigated plants (Table S5), while the Braw genotype showed declining trends in all physicochemical measures except TA under the second stress stage. Sandra had the largest increasing percentages in TSS (1.75%), CAC (9.47%), and SSC (11.81%) with SOBS application, followed by Raza Pashayi in ASC (28.71%) and TPC (27.20%) in the presence of SOES application during both stress stages (Table S5).

**Table 4.** Effect of tomato genotypes treated at various stress stages with oak leaf powder, oak leaf extract, and biofertilizer on the fruit physicochemical traits. Increasing and decreasing percentages are indicated by positive and negative values, respectively.

| | Increasing and Decreasing Percentages Compared with Irrigated Plants during the First Stress Stage | | | | | | |
|---|---|---|---|---|---|---|---|
| **Genotypes** | **MC (%)** | **TA (%)** | **TSS (%)** | **ASC (%)** | **CAC (%)** | **SSC (%)** | **TPC (%)** |
| Raza Pashayi | −0.45 a ± 0.14 | −1.49 b ± 9.61 | −2.06 b ± 2.35 | 29.29 a ± 6.52 | −0.84 b ± 1.37 | 5.73 b ± 3.59 | 19.82 a ± 2.46 |
| Sandra | −0.35 a ± 0.18 | 5.97 a ± 5.27 | 1.32 a ± 2.84 | 25.46 b ± 10.82 | 2.22 a ± 0.73 | 9.07 a ± 3.84 | 14.18 b ± 2.72 |
| Braw | −1.46 c ± 0.68 | 7.46 a ± 6.65 | −6.68 c ± 2.81 | 0.26 d ± 5.12 | −8.24 c ± 3.86 | −9.45 c ± 8.47 | −8.43 d ± 3.12 |
| Yadgar | −1.14 b ± 0.25 | 8.77 a ± 5.84 | −7.49 c ± 3.07 | 12.93 c ± 6.40 | −10.95 d ± 3.73 | −9.94 c ± 4.69 | 8.20 c ± 1.97 |
| | Increasing and Decreasing Percentages Compared with Irrigated Plants during the Second Stress Stage | | | | | | |
| **Genotypes** | **MC (%)** | **TA (%)** | **TSS (%)** | **ASC (%)** | **CAC (%)** | **SSC (%)** | **TPC (%)** |
| Raza Pashayi | −0.39 a ± 0.20 | −3.71 c ± 5.59 | −1.93 b ± 1.54 | 24.72 b ± 4.60 | 2.04 b ± 1.51 | 2.73 b ± 2.79 | 24.29 a ± 6.83 |
| Sandra | −0.33 a ± 0.17 | 6.01 b ± 6.63 | 1.32 a ± 2.09 | 27.38 a ± 12.61 | 5.01 a ± 1.03 | 12.67 a ± 4.54 | 15.19 b ± 4.13 |
| Braw | −1.44 c ± 0.57 | 14.87 a ± 7.93 | −6.15 c ± 3.28 | −11.16 d ± 2.62 | −13.05 d ± 2.80 | −15.41 d ± 7.74 | −8.02 d ± 1.74 |
| Yadgar | −1.14 b ± 0.29 | 7.03 b ± 5.53 | −7.97 d ± 3.12 | 10.46 c ± 8.75 | −8.92 c ± 7.82 | −6.64 c ± 4.02 | 6.18 c ± 1.03 |
| | Increasing and Decreasing Percentages Compared with Irrigated Plants during the First and Second Stress Stages | | | | | | |
| **Genotypes** | **MC (%)** | **TA (%)** | **TSS (%)** | **ASC (%)** | **CAC (%)** | **SSC (%)** | **TPC (%)** |
| Raza Pashayi | −0.42 a ± 0.20 | 3.12 c ± 5.58 | −2.90 b ± 3.32 | 20.78 a ± 7.76 | 0.19 b ± 2.75 | 0.33 b ± 3.46 | 21.28 a ± 6.70 |
| Sandra | −0.65 b ± 0.14 | 9.65 b ± 5.28 | 0.24 a ± 2.71 | 20.57 a ± 9.39 | 6.95 a ± 1.89 | 7.81 a ± 4.09 | 12.68 b ± 3.67 |
| Braw | −2.14 d ± 0.70 | 17.57 a ± 10.89 | −8.79 c ± 1.59 | −17.15 c ± 6.30 | −16.17 c ± 4.31 | −18.87 d ± 7.90 | −11.39 d ± 2.23 |
| Yadgar | −1.31 c ± 0.32 | 14.01 a ± 5.66 | −11.09 d ± 2.59 | 4.96 b ± 6.22 | −15.68 c ± 2.75 | −11.17 c ± 3.38 | 6.39 c ± 1.82 |

MC: moisture content, TA: titratable acidity, TSS: total soluble solids, ASC: ascorbic acid content, CAC: carotenoid content, SSC: soluble sugar content, TPC: total phenolics content. Any mean values sharing the same letter in the same column are not statistically significant, as determined by the Duncan's multiple range test at $p \leq 0.05$. The value is represented by trait index ± standard deviation (SD). Each value is the average of three measurements.

### 3.3. Impact of Various Treatments on the Biochemical Responses of the Leaves of Tomato Plants under Conditions of Water Stress

To gain a better understanding of the mechanism of tolerance in plants treated with SS, SOS, SOES, and SOBS under water deficit stress, a number of biochemical measurements were performed on the leaves of tomato plants. As shown in Table S3, significant variations were detected among different levels of treatment for all biochemical characters of the leaves of the tomato under all stress stages. The maximum values of proline content (PC), soluble sugar content (SSC), guaiacol peroxidase (GPA), and catalase (CAT) were recorded by the tomato plants treated with SOES, while the highest values of total phenolic content

(TPC) and antioxidant activity (AC) were observed by the plants treated with SOBS under the first and second stress stages. Moreover, under the combination of first and second stress stages, the plants treated with SOBES displayed the greatest values of all biochemical traits, with the exception of the LP trait. Furthermore, the control plants (SW) exhibited the minimum values of all chemical characters of the leaves of tomato under all stress stages. Low amounts of lipid peroxidation were observed by SW (5.24 nmol g$^{-1}$ FLW), followed by SOES (7.15 nmol g$^{-1}$ FLW) and SOBS (8.46 nmol g$^{-1}$ FLW), under the first, second, and their combination stress stages (Table 5).

Seven different variables relating to the biochemical parameters of leaves treated with SW, SS, SOS, SOES, and SOBS were subjected to a principal component analysis (PCA). Based on an eigenvalue greater than one, the first two components displayed cumulative distributions of 93.53, 93.35, and 98.11% for the first, second, and their combined stress stages, respectively (Figure 2A–C). The biochemical characteristics and treatments were dispersed in various ways across the PCA plot throughout the first, second, and combined stages of stress. The characteristics that had the most significance in affecting the observed variance along PC1 were PC, SSC, TPC, AC, GPA, and CAT. However, the LP characteristic was the primary driver of variance along PC2. In comparison with untreated plants (SS), the application of SOBS, SOES, and SOS reduced the amount of lipid peroxidation in the leaves during all stages of stress. On the right side (blue circle) of the PCA plot, characteristics of SOBS- and SOES-treated plants with high PC, TPC, AC, SSC, GPA, and CAT values were noted throughout the first, second, and their combined stress stages. On the other hand, the plants that received SS treatment produced more LP (brown circle).

**Table 5.** Impact of oak leaf powder, oak leaf extract, and biofertilizer on the biochemical characteristics of the leaves of tomato plants under various stress stages.

| Treatment | PC (µg g$^{-1}$) | SSC (µg g$^{-1}$) | TPC (µg g$^{-1}$) | AC (µg g$^{-1}$) | LP (nmol g$^{-1}$) | GPA (units min$^{-1}$ g$^{-1}$) | CAT (units min$^{-1}$ g$^{-1}$) |
|---|---|---|---|---|---|---|---|
| **First Stress Stage** | | | | | | | |
| SOBS | 1546.37 b ± 503.08 | 569.04 b ± 99.21 | 433.90 a ± 98.38 | 1010.20 a ± 173.44 | 8.46 c ± 1.13 | 0.26 b ± 0.06 | 139.61 b ± 42.49 |
| SOES | 1956.50 a ± 489.76 | 612.64 a ± 109.34 | 399.21 b ± 90.59 | 1006.99 b ± 175.26 | 7.15 d ± 0.98 | 0.34 a ± 0.06 | 160.71 a ± 56.00 |
| SOS | 1322.91 c ± 619.10 | 524.14 c ± 96.68 | 344.91 c ± 57.07 | 966.79 c ± 171.26 | 11.05 b ± 2.24 | 0.25 b ± 0.08 | 118.51 c ± 65.65 |
| SS | 1307.65 d ± 578.09 | 417.19 d ± 108.74 | 325.57 d ± 56.09 | 892.80 d ± 94.45 | 13.10 a ± 2.26 | 0.16 c ± 0.09 | 87.66 d ± 45.71 |
| SW | 1054.58 e ± 425.20 | 374.14 e ± 91.47 | 312.23 e ± 63.52 | 893.31 d ± 129.46 | 5.24 e ± 0.78 | 0.13 d ± 0.06 | 64.94 e ± 42.26 |
| **Second Stress Stage** | | | | | | | |
| SOBS | 2058.81 b ± 426.81 | 742.65 b ± 110.77 | 428.24 a ± 20.91 | 986.05 a ± 120.82 | 9.92 c ± 1.40 | 0.24 b ± 0.08 | 126.62 b ± 30.89 |
| SOES | 2534.00 a ± 433.44 | 782.93 a ± 89.71 | 402.68 b ± 29.48 | 974.43 b ± 159.49 | 8.17 d ± 1.68 | 0.33 a ± 0.08 | 159.09 a ± 40.27 |
| SOS | 1813.81 c ± 396.27 | 627.76 c ± 146.44 | 378.99 c ± 35.48 | 909.19 c ± 163.83 | 10.59 b ± 1.32 | 0.23 c ± 0.06 | 113.64 c ± 24.69 |
| SS | 1616.82 d ± 444.00 | 529.00 d ± 122.78 | 322.51 e ± 70.59 | 902.40 d ± 139.83 | 12.83 a ± 2.85 | 0.17 d ± 0.07 | 81.17 d ± 15.04 |
| SW | 1126.37 e ± 533.06 | 501.76 e ± 145.17 | 334.84 d ± 32.40 | 895.51 e ± 109.70 | 7.20 e ± 0.51 | 0.15 e ± 0.08 | 64.94 e ± 40.50 |
| **Combination of Both Stress Stages** | | | | | | | |
| SOBS | 2057.01 b ± 391.73 | 764.63 b ± 121.99 | 453.15 b ± 58.23 | 1029.29 b ± 96.55 | 10.67 c ± 1.11 | 0.30 b ± 0.11 | 155.84 b ± 63.92 |
| SOES | 2217.65 a ± 330.37 | 856.54 a ± 96.24 | 493.69 a ± 122.67 | 1092.47 a ± 120.92 | 8.94 d ± 1.21 | 0.44 a ± 0.19 | 217.53 a ± 90.38 |
| SOS | 1689.96 c ± 485.67 | 670.35 c ± 109.86 | 407.97 c ± 59.26 | 938.58 c ± 96.75 | 12.53 b ± 2.66 | 0.27 c ± 0.10 | 137.99 c ± 20.56 |
| SS | 1661.24 d ± 268.52 | 580.69 d ± 12.76 | 316.07 e ± 76.85 | 907.34 d ± 118.09 | 13.95 a ± 2.80 | 0.17 d ± 0.09 | 95.78 d ± 17.32 |
| SW | 1126.37 e ± 533.06 | 501.76 e ± 145.17 | 334.84 d ± 32.40 | 895.51 d ± 109.70 | 7.20 e ± 0.51 | 0.15 e ± 0.08 | 64.94 e ± 40.50 |

PC: proline content, SSC: soluble sugar content, TPC: total phenolic content, AC: antioxidant activity, LP: lipid peroxidation, GPA: peroxidase, CAT: catalase, SS: stressed plants that had not been treated, SOS: stressed plants that had been treated with oak leaf powder, SOES: stressed plants that had been treated with oak leaf powder and oak leaf extract, SOBS: stressed plants that had been treated with oak leaf powder and biofertilizers. Duncan's multiple range test at $p \leq 0.05$ indicates that any mean values sharing the same letter in the same column are not statistically significant. The value is represented by mean ± standard deviation (SD). Each value is the average of three measurements.

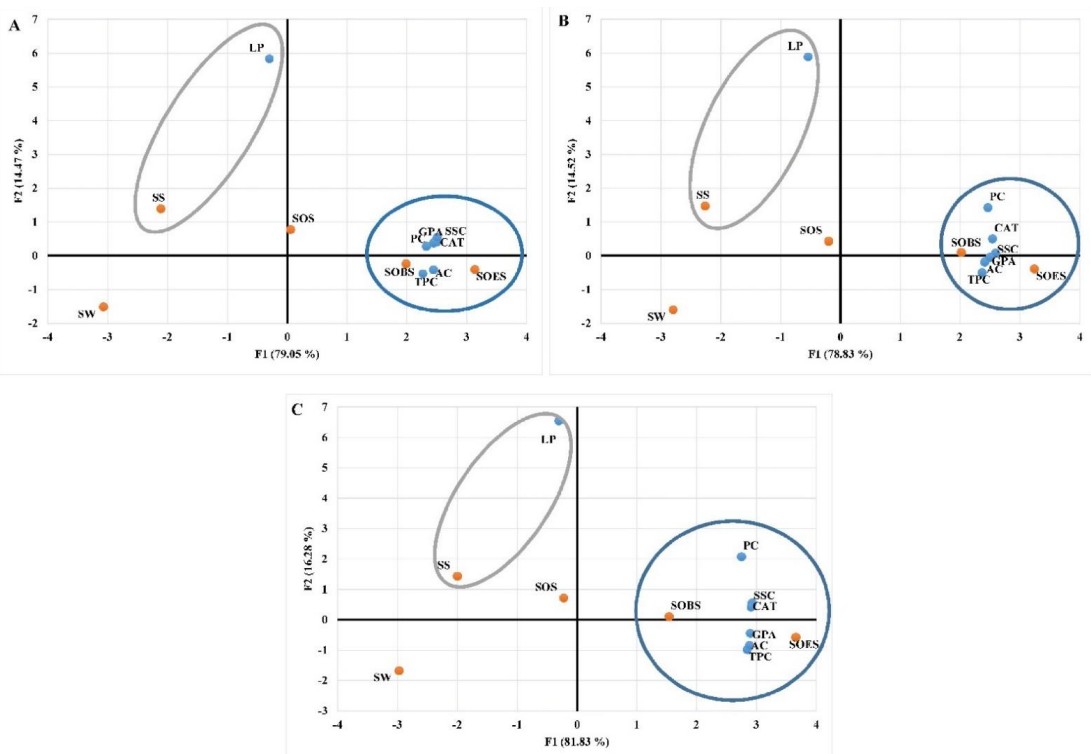

**Figure 2.** PCA plot illustrating the distribution of leaf biochemical characteristics and treatments under first (**A**), second (**B**), and both (**C**) stress circumstances. PC: proline content, SSC: soluble sugar content, TPC: total phenolic content, AC: antioxidant activity, LP: lipid peroxidation, GPA: peroxidase, CAT: catalase, SS: stressed plants that had not been treated, SOS: stressed plants that had been treated with oak leaf powder, SOES: stressed plants that had been treated with oak leaf powder and oak leaf extract, SOBS: stressed plants that had been treated with oak leaf powder and biofertilizers. F1 and F2 represent the first and second components, respectively.

### 3.4. Impact of Different Genotypes Treated with SW, SS, SOS, SOES, and SOBS on the Biochemical Responses of the Leaves of Tomato Plants under Circumstances of Water Stress

Tomato plant leaves were analyzed chemically in order to acquire a better knowledge of the mechanism of tolerance in genotypes treated with SS, SOS, SOES, and SOBS. As demonstrated in Table S3, substantial differences were identified between different genotypes for all biochemical characteristics of tomato leaves under all stress stages. The tolerant genotype Sandra had the highest values of PC, SSC, and AC during the first stress stage, whereas the tolerant genotype Raza Pashayi had the highest scores of TPC, GPA, and CAT traits. The sensitive genotype Yadgar showed the minimum values of all chemical characteristics with the exception of the LP trait. As a comparison between tolerant and sensitive genotypes, the mean values of SSC, GPA, and CAT in tolerant genotypes were higher than those obtained in sensitive plants. The highest scores of LP were found in sensitive plants (Table 6). Under the second stress stage, the tolerant genotype Sandra had the highest values of PC, TPC, AC, and CAT, while the tolerant genotype Raza Pashayi had the highest value of GPA. Except for the PC and LP features, the sensitive genotype Yadgar displayed the lowest values for all biochemical parameters (Table 6). Comparing tolerant and sensitive genotypes, the mean TPC, AC, and GPA values of tolerant genotypes were greater than those of sensitive plants. The susceptible plants (Braw and Yadgar) had the highest levels of LP. Sandra's genotype exhibited the greatest PC, AC, and CAT scores in response to both stress periods. With the exception of the LP trait, Yadgar genotypes had the lowest values for all leaf biochemical parameters (Table 6).

**Table 6.** Effects of oak leaf powder, oak leaf extract, and biofertilizer on the biochemical traits of the leaves of tomato plants under different levels of stress.

| | | | First Stress Stage | | | | |
|---|---|---|---|---|---|---|---|
| Genotypes | PC ($\mu$g g$^{-1}$) | SSC ($\mu$g g$^{-1}$) | TPC ($\mu$g g$^{-1}$) | AC ($\mu$g g$^{-1}$) | LP (nmol g$^{-1}$) | GPA (units min$^{-1}$ g$^{-1}$) | CAT (units min$^{-1}$ g$^{-1}$) |
| Raza Pashayi | 1093.08 d $\pm$ 146.59 | 517.72 b $\pm$ 160.20 | 433.82 a $\pm$ 65.06 | 987.30 c $\pm$ 34.97 | 6.86 d $\pm$ 2.03 | 0.30 a $\pm$ 0.06 | 172.73 a $\pm$ 40.01 |
| Sandra | 2220.97 a $\pm$ 257.08 | 610.25 a $\pm$ 88.53 | 371.72 c $\pm$ 70.52 | 1080.95 a $\pm$ 106.77 | 9.39 b $\pm$ 2.34 | 0.26 b $\pm$ 0.12 | 132.47 b $\pm$ 46.10 |
| Braw | 1252.82 b $\pm$ 497.38 | 499.26 c $\pm$ 75.14 | 393.63 b $\pm$ 48.06 | 1020.41 b $\pm$ 103.02 | 9.28 c $\pm$ 3.50 | 0.19 c $\pm$ 0.07 | 93.51 c $\pm$ 59.81 |
| Yadgar | 1183.54 c $\pm$ 526.99 | 370.49 d $\pm$ 69.11 | 253.48 d $\pm$ 21.77 | 727.43 d $\pm$ 34.97 | 10.46 a $\pm$ 3.77 | 0.16 d $\pm$ 0.08 | 58.44 d $\pm$ 17.35 |
| | | | Second Stress Stage | | | | |
| Genotypes | PC ($\mu$g g$^{-1}$) | SSC ($\mu$g g$^{-1}$) | TPC ($\mu$g g$^{-1}$) | AC ($\mu$g g$^{-1}$) | LP (nmol g$^{-1}$) | GPA (units min$^{-1}$ g$^{-1}$) | CAT (units min$^{-1}$ g$^{-1}$) |
| Raza Pashayi | 1620.41 c $\pm$ 526.17 | 606.11 c $\pm$ 181.90 | 378.73 b $\pm$ 51.11 | 1010.00 b $\pm$ 59.28 | 8.93 c $\pm$ 1.88 | 0.31 a $\pm$ 0.06 | 103.90 bc $\pm$ 27.77 |
| Sandra | 2278.41 a $\pm$ 491.93 | 658.77 b $\pm$ 100.11 | 414.37 a $\pm$ 27.83 | 1102.97 a $\pm$ 47.37 | 7.93 d $\pm$ 1.19 | 0.23 b $\pm$ 0.12 | 128.57 a $\pm$ 68.90 |
| Braw | 1621.13 c $\pm$ 938.94 | 795.56 a $\pm$ 82.71 | 373.60 c $\pm$ 23.28 | 858.92 c $\pm$ 27.47 | 10.77 b $\pm$ 2.48 | 0.19 c $\pm$ 0.07 | 106.49 b $\pm$ 47.70 |
| Yadgar | 1799.90 b $\pm$ 121.84 | 486.85 d $\pm$ 110.53 | 327.12 d $\pm$ 74.40 | 762.16 d $\pm$ 54.98 | 11.34 a $\pm$ 3.01 | 0.17 d $\pm$ 0.04 | 97.40 c $\pm$ 18.69 |
| | | | Combination of Both Stress Stages | | | | |
| Genotypes | PC ($\mu$g g$^{-1}$) | SSC ($\mu$g g$^{-1}$) | TPC ($\mu$g g$^{-1}$) | AC ($\mu$g g$^{-1}$) | LP (nmol g$^{-1}$) | GPA (units min$^{-1}$ g$^{-1}$) | CAT (units min$^{-1}$ g$^{-1}$) |
| Raza Pashayi | 1903.64 b $\pm$ 567.18 | 658.83 c $\pm$ 190.89 | 432.28 b $\pm$ 97.33 | 1003.38 b $\pm$ 57.89 | 9.30 c $\pm$ 2.13 | 0.40 a $\pm$ 0.13 | 132.47 b $\pm$ 47.07 |
| Sandra | 2114.72 a $\pm$ 403.89 | 724.07 b $\pm$ 157.46 | 405.43 c $\pm$ 18.45 | 1082.30 a $\pm$ 52.67 | 9.01 d $\pm$ 1.28 | 0.29 b $\pm$ 0.21 | 176.62 a $\pm$ 125.44 |
| Braw | 1382.77 d $\pm$ 117.24 | 793.94 a $\pm$ 72.50 | 459.33 a $\pm$ 118.99 | 955.36 c $\pm$ 168.75 | 12.48 a $\pm$ 3.63 | 0.17 d $\pm$ 0.06 | 103.90 c $\pm$ 36.73 |
| Yadgar | 1600.67 c $\pm$ 669.52 | 522.35 d $\pm$ 126.03 | 307.53 d $\pm$ 65.85 | 849.50 d $\pm$ 79.51 | 11.85 b $\pm$ 3.14 | 0.20 c $\pm$ 0.06 | 124.68 b $\pm$ 30.28 |

PC: proline content, SSC: soluble sugar content, TPC: total phenolic content, AC: antioxidant activity, LP: lipid peroxidation, GPA: peroxidase, CAT: catalase. Duncan's multiple range test at $p \leq 0.05$ reveals that any mean values in the same column that share the same letter are not statistically significant. The value is represented by mean $\pm$ standard deviation (SD). Each value is the average of three measurements.

Raza Pashayi had the highest values in SSC (711.79 $\mu$g g$^{-1}$), TPC (518.13 $\mu$g g$^{-1}$), and CAT (220.78 units min$^{-1}$ g$^{-1}$) in the availability of the SOES treatment, while Sandra had the highest values in PC (2446.05 $\mu$g g$^{-1}$) and GPA (0.42 units min$^{-1}$ g$^{-1}$) under the first stress stage, as shown in Table S6. In comparison with irrigated plants during the second stress stage, the Sandra genotype recorded the highest values for AC (1151.89 $\mu$g g$^{-1}$), GPA (0.40 units min$^{-1}$ g$^{-1}$), and CAT (214.29 units min$^{-1}$ g$^{-1}$). When SOES was applied, Sandra had the highest SSC (976.60 $\mu$g g$^{-1}$), GPA (0.62 units min$^{-1}$ g$^{-1}$), and CAT (363.64 units min$^{-1}$ g$^{-1}$) scores, while Raza Pashayi had the highest PC (2571.69 $\mu$g g$^{-1}$) score during both stress stages.

### 3.5. GC/MS Analysis of Oak Leaf Extract

Table 7 displays the phytochemical composition of the extracts as determined by GC/MS analysis. The extract contained twenty-four components. The major compounds were heptasiloxane, 1,1,3,3,5,5,7,7,9,9,11,11,13,13-tetradecamethyl-(32.50%), silane, dimethoxydimethyl-(11.67), octasiloxane, 1,1,3,3,5,5,7,7,9,9,11,11,13,13,15,15-hexadecamethyl-(10.88%), 1-hexadecanol (9.37%), behenic alcohol (8.86%), 2,4-di-tert-butylphenol (7.02%), 1-octadecene (6.73%), acetic acid, chloro-, octadecyl ester (1.55%), dichloroacetic acid, 4-hexadecyl ester (1.52%), coumatetralyl isomer-2 ME (1.49%), 1-dodecanol (1.29%), chloroacetic acid, and pentadecyl ester (1.01%).

**Table 7.** Substances detected by GC/MS analysis and biological activity of the major compounds in the leaf water extract of *Quercus aegilops*.

| Name of Compound | Retention Time (min) | Peak Area | Concentration (%) | Biological Activity of Major Compounds |
|---|---|---|---|---|
| Silane, dimethoxydimethyl- | 5.17 | 7,202,705.00 | 11.67 | Antibacterial [43] |
| Cyclotrisiloxane, hexamethyl- | 6.93 | 327,535.00 | 0.53 | |
| Silane, methyldimethoxyethoxy- | 8.30 | 268,638.00 | 0.44 | |
| Oxime-, methoxy-phenyl- | 9.38 | 231,616.00 | 0.38 | |
| Tetraethyl silicate | 10.54 | 314,412.00 | 0.51 | |
| 1-Dodecanol | 13.90 | 796,766.00 | 1.29 | Antibacterial [44] |
| 1-Hexadecanol | 16.73 | 1,941,423.00 | 9.37 | Reduction of evaporation [45] |
| Carbonic acid, decyl undecyl ester | 16.84 | 397,832.00 | 0.64 | |
| 7-Tetradecene | 16.90 | 307,446.00 | 0.50 | |

**Table 7.** *Cont.*

| Name of Compound | Retention Time (min) | Peak Area | Concentration (%) | Biological Activity of Major Compounds |
|---|---|---|---|---|
| Chloroacetic acid, tetradecyl ester | 17.04 | 250,824.00 | 0.41 | |
| 2,4-Di-tert-butylphenol | 18.29 | 4,329,760.00 | 7.02 | Antioxidant [46–48] |
| Carbonic acid, eicosyl vinyl ester | 19.31 | 422,435.00 | 0.68 | |
| Dichloroacetic acid, 4-hexadecyl ester | 19.36 | 536,815.00 | 1.52 | Antimicrobial [49] |
| 1-Octadecene | 21.45 | 4,150,402.00 | 6.73 | Antioxidant and antimicrobial [50,51] |
| Acetic acid, chloro-, octadecyl ester | 21.58 | 542,636.00 | 1.55 | No activity was reported |
| 1,2-Benzenedicarboxylic acid, bis(2-methylpropyl) ester | 22.32 | 229,532.00 | 0.37 | |
| 18-Norabietane | 23.10 | 242,753.00 | 0.39 | |
| Behenic alcohol | 23.48 | 3,132,644.00 | 8.86 | Antifungal [52] |
| Chloroacetic acid, pentadecyl ester | 23.58 | 273,527.00 | 1.01 | No activity was reported |
| Coumatetralyl isomer-2 ME | 23.67 | 918,610.00 | 1.49 | No activity was reported |
| Acetic acid, chloro-, octadecyl ester | 24.34 | 507,902.00 | 0.82 | |
| Cyclotetrasiloxane, octamethyl- | 27.02 | 268,576.00 | 0.44 | |
| Heptasiloxane, 1,1,3,3,5,5,7,7,9,9,11,11,13,13-tetradecamethyl- | 32.59 | 21,294,993.00 | 32.50 | Insecticidal and antibacterial [53,54] |
| Octasiloxane, 1,1,3,3,5,5,7,7,9,9,11,11,13,13,15,15-hexadecamethyl- | 38.79 | 6,719,421.00 | 10.88 | |

## 4. Discussion

Plant growth results from cell division, cell enlargement, and differentiation and is regulated by a wide range of genetic, physiological, ecological, and morphological processes, as well as the interaction between these factors [42]. Damage to physiological and biochemical processes, such as a delay in stomatal conductance, a decrease in nutrient uptake, a breakdown of leaf pigments, a decrease in photosynthesis, a stop in the rate of net assimilation and photosystem photochemical efficiency parameters, an increase in reactive oxygen species (ROS), and oxidative damage caused by water stress, reduced the morphological features [55]. The fresh weight, plant height, and productivity of the stressed tomato plants were all lower than those of the control plant (watered plants), as found by previous studies [12,56–58]. Relative water content and total chlorophyll content also decreased under SS condition. The same results were also found in tomato plants studied by Khan et al. [59], Ibrahim et al. [55], and Ullah et al. [42].

Root fresh weight (RFW) and root dry weight (RDW) under situations of water stress have shown significantly increased percentages for all degrees of treatment under all stress stages. The plant treated with SOBS and SOES had significantly higher RFW and RDW trait values than the control group (SW) during all stress stages. As a comparison among the three stress stages, the plants treated with SOBS showed the greatest increases in RWF (107%) and RDW (127.80%) in the second stress stage. The increased root surface area and root volume in plants during the search for water in the soil is mostly responsible for the higher RFW and RDW observed across all stress stages in comparison with untreated and unstressed plants. Additionally, a large number of prominent compounds found in leaf extract, including silane, heptasiloxane, and octasiloxane are thought to be silicon (Si) sources and are responsible for the increasing RL, RFW, and RDW in plants exposed to SOES at all stress levels. The leaf extract also had the compounds 2,4-Di-tert-butylphenol and 1-octadecene, which have antioxidant properties that reduce the synthesis of ROS products and membrane lipid peroxidation [46–48,50]. In addition, the leaf extract contained a 1-hexadecanol compound, which is used to reduce water evaporation in reservoirs. Si-enhanced cell-wall extensibility in the root's growth zone likely contributes to root elongation. Root density and length were both increased by Si in Purslane [60]. Sorghum's root length was found to be increased by Si, according to research by Sonobe et al. [61]. It is also likely that the higher RFW and RDW in SOES-treated plants are due to the ability of Si and 2,4-Di-tert-butylphenols to minimize ROS overproduction, which reduces membrane

lipid peroxidation. On the other hand, our research showed that both SFW and SDW were lower in the SOES-treated plant. This may be because of the fact that Si controls the levels of polyamine and 1-aminocyclopropane-1-carboxylic acid in response to drought stress, which improves root growth, the ratio of roots to shoots, water uptake at the roots, and hydraulic conductance. Root endodermal silicification and suberization are also boosted by Si-mediated alterations in root growth, which help plants better retain water and tolerate the negative effects of drought [62]. In comparison with plants treated with SS and SOS, SFW and SDW in plants treated with SOES and SOBS may have increased owing to a decrease in ROS products and membrane lipid peroxidation. Furthermore, RFW and RDW increased in plants treated with SOBS throughout all stress stages, and these increases were induced by the presence of cytokinin, enzymes (lipase, amylase, protease, and chitinase), *Bacillus subtilis*, and *Pesudomonas putida*. These components of the SOBS treatment improve root area and volume by degrading organic matter and boosting phosphorus availability in the soil [63]. *Bacillus subtilis* and *Pesudomonas putida* invade plant rhizospheres and produce volatile organic chemicals that can affect plant development and root architecture in a variety of plants [64,65].

Drought stress, on the other hand, can alter the chemical composition of fruits. Organic acids (malic and citric acid) and soluble sugars are among the primary osmotic components found in ripe fruits [66]. Organic acids are stored by plants in order to reduce their osmotic potential and prevent cell turgor pressure from decreasing [67,68]. Vitamin C, also known as ascorbic acid, is found in all parts of plants. It plays a pivotal role in the development and expansion of plants. Ascorbic acid is the plant's primary antioxidant, which neutralizes the active forms of oxygen. Our results showed that the ascorbic acid content of the red fruit of the stressed plant increased owing to the water shortage. This increase in ascorbic acid may be vital for detoxifying reactive oxygen species. Antioxidant capability is determined by the phenolic contents of tomato fruits (TPC), and an increase in the TPC amount results in a decrease in oxidative alterations in cells owing to a lower concentration of free radicals [69,70].

Fructose and glucose levels both increase sharply when tomatoes ripen. The total soluble solids (TSS) concentration is influenced by the carbohydrate, organic acid, protein, fat, and mineral components. Our results suggest that shifts in the glucose/fructose ratio and organic acid levels may be responsible for the observed reduction in TSS in our investigation [66]. Compared with SS and SOS circumstances, the availability of silane, heptasiloxane, octasiloxane, and 2,4-Di-tert-butylphenol increases SSC during SOES application, which decreases ROS production by triggering antioxidant systems.

The results of the genotype effects revealed that tomato genotypes responded differently to SS, SOS, SOES, and SOBS applications under water stress. According to ASC, CAC, SSC, and TPC data, drought stress reduced the quality of tomato tolerant genotypes treated with SOS, SOES, and SOBS.

Different reactions were seen in terms of the leaf biochemical responses in plants treated with SS, SOS, SOES, and SOBS under stressful conditions. The highest levels of lipid peroxidation (LP), a metabolic process that results in the oxidative degradation of lipids by reactive oxygen species, were observed in the untreated and stressed genotype condition (SS). As a result of this process, the lipids in the cell membrane may break down, which can damage the cell and lead to its death. Low accumulations of biochemical compounds such as TPC, PC, SSC, AC, GPA, and CAT are responsible for this increase in LP. The genotypes treated under SOES and SOBS conditions, on the other hand, showed the highest levels of TPC, PC, SSC, AC, GPA, and CAT, which led to the reduction of LP. Furthermore, the SOES application may have induced the antioxidant systems, which may have contributed to the availability of silane, heptasiloxane, octasiloxane, and 2,4-Di-tert-butylphenol in the leaf extract.

Different responses were observed for the tolerant and sensitive genotypes during stress stages. Owing to the low accumulation of SSC, PC, TPC, AC, GPA, and CAT in sensitive geometries, the findings of leaf biochemical parameters showed the maximum

LP. Different response profiles between the tolerant genotypes were found. Under the first stages of stress, Raza Pashayi demonstrated the highest levels of TPC, GPA, and CAT, whereas Sandra's genotype had the highest levels of SSC, PC, and AC. Raza Pashyi recorded the highest values for GPA, AC, and SSC traits during the second stress stage, while Sandra had the maximum values for TPC, CAT, and AC.

## 5. Conclusions

According to our findings, the genotypes responded differently to the application of SS, SOS, SOES, and SOBS at various stress stages. In contrast to untreated and stressed plants, tomato plants treated with SOS, SOES, and SOBS showed a slight decrease in the morpho-physiological and fruit physicochemical attributes in response to drought stress. Additionally, a combination of the two stress stages resulted in a greater decrease in these features than either the first or second stress stage alone. All tomato genotypes exposed to SOES and SOBS exhibited significant levels of TPC, ASC, and SSC characteristics along with low amounts of TA in fruit. In fruit TPC, ASC, TSS, CAC, and SSC, the in vitro tolerant genotypes (Sandra and Raza Pashyi) outperformed the in vitro intolerant genotypes (Braw and Yadgar). In the leaf tolerant genotypes treated with SOES and SOBS, the lowest levels of lipid peroxidation and the highest levels of TPC, AC, SSC, PC, GPA, and CAT were found. Based on the findings of this study, Raza Pashyi and Sandra are ideal for growing in places with limited water availability. Furthermore, these genotypes are beneficial for breeding projects aimed at developing drought-tolerant tomato cultivars. Furthermore, the use of oak leaf powder, oak leaf extract, and biofertilizer reduced the effect of drought stress on tomato plants. However, the use of oak leaf powder and oak leaf extract can be described as novel agricultural practices because they are low-cost, simple to use, and time-consuming, and they can meet the growing demands of the agricultural sector by providing environmentally sustainable techniques for enhancing plant resistance to abiotic stress. The usage of the combination of leaf crude extract, oak leaf powder, and arbuscular mycorrhizal fungus should be investigated further under stress conditions. In order to determine the biostimulation effects of oak leaf powder and oak leaf extract, it is important to test their impacts on plant growth and production under normal conditions.

**Supplementary Materials:** The following supporting information can be downloaded at https://www.mdpi.com/article/10.3390/agriculture12122082/s1, Table S1. F and probability (P) values of different morpho-physiological traits of tomato genotypes treated with different treatments under the first, second, and their combinations of stress stages; Table S2. F and probability (P) values of different fruit physicochemical characters of tomato genotypes treated with various treatments under the first, second, and their combinations of stress stages; Table S3. F and probability (P) values of different leaf biochemical parameters of tomato genotypes treated with different treatments under the first, second, and their combinations of stress stages; Table S4. Interaction effects of tomato genotypes and treatments on the different morpho-physiological characters under the first, second, and their combinations of stress stages; Table S5. Interaction effects of tomato genotypes and treatments on the different physicochemical traits under the first, second, and their combinations of stress stages; Table S6. Interaction effects of tomato genotypes and treatments on the different biochemical traits under the first, second, and their combination of stress stages; Figure S1. Experimental design layout of the variables investigated in this study for each genotype; Figure S2. Effect of different treatments on the root morphology of the Yadgar genotype under different stress conditions; Figure S3. Effect of different treatments on the fruit morphology of the Raza Pashayi genotype under first stress conditions.

**Author Contributions:** Conceptualization, N.A.-r.T. and F.M.W.G.; methodology, K.S.R. and D.D.L.; data curation, D.D.L. and K.S.R.; formal analysis, N.A.-r.T.; writing—original draft preparation, N.A.-r.T. and K.S.R.; writing—review and editing, N.A.-r.T., K.S.R. and D.D.L. All authors have read and agreed to the published version of the manuscript.

**Funding:** This research received no external funding.

**Institutional Review Board Statement:** Not applicable.

**Informed Consent Statement:** Not applicable.

**Data Availability Statement:** The article and supplementary files contain all data.

**Acknowledgments:** The authors would like to thank the University of Sulaimani College of Agricultural Engineering Sciences staff for their help and assistance during this project.

**Conflicts of Interest:** The authors declare no conflict of interest.

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
