# Peer review of "Effects of Oak Leaf Extract, Biofertilizer, and Soil Containing Oak Leaf Powder on Tomato Growth and Biochemical Characteristics under Water Stress Conditions"

_agriculture, doi:10.3390/agriculture12122082_

Round 1

Reviewer 1 Report

The manuscript entitled Effects of Oak Leaf Extract, Biofertilizer, and Soil Containing Oak Leaf Powder on Tomato Growth and Biochemical Characteristics Under Water Stress Conditions" is based on original research experiment and the presented results therein broaden the knowledge of plant biology and agronomy. The aims of this work was to determine the effects of oak leaf extract, biofertilizer, and soil incorporating oak leaf powder on the growth and biochemical traits of four tomato genotypes under water stress conditions during two stages of plant development. The object of work were two susceptible tomato genotypes, Braw and Yadgar, and two tolerant tomato genotypes, Raza Pashayi and Sandra. Authors performed experiment in controled conditions, during which analysis of morphological and physiological parameters  was performed.

There is no doubt that this work is in the scope of Agriculture. The publication presents some interesting studies. The work delivers some interesting results and can be important source of valuable information.

The introduction is generally properly composed. The materials and methods section contains the basic requested elements and provide information about the experimental preparations and analyses. The data analysis is properly provided. The results show valuable information. The obtained data are discussed sufficiently.

However, the authors made some shortcomings that should be corrected before the publication of the work:

1)      Abstract: the information why biostimulants are so important should be added. On the other side, the results are to descriptive in this part.

2) Authors did not make a research hypothesis. However, the aims of the study is clearly stated at the end of the introduction section.  

3) MM section: the information about conditions during experiments should be described: what type of soil was used, by which tool the moisture of soils was measured, what was the temperature and air humidity?

4) MM section: the statistical procedure should be described more detailed: what was p value, what was the number of repetition? Moreover, in figure and table captions should be explained what symbols ± means (SE or SD?).

5) In conclusion there should mechanistic response/ description what the investigated mechanism looks like. 

I would like to underline that my remarks are auxiliary and not undertake the quality and importance of the paper.

Author Response

Manuscript number: agriculture- 2072747

Paper title: Effects of Oak Leaf Extract, Biofertilizer, and Soil Containing Oak Leaf Powder on Tomato Growth and Biochemical Characteristics Under Water Stress Conditions

Authors: Nawroz Abdul-razzak Tahir, Kamaran Salh Rasul, Djshwar Dhahir Lateef, Florian M. W. Grundler

Dear Editor and Reviewer

The authors would like to thank the area editor and the reviewers for their precious time and invaluable comments. We have carefully addressed all the comments. The constructive comments/ suggestions by the reviewer are really appreciated. We have now completely revised the manuscript. All corrections in English language and updating of information are highlighted by the red lines. The corresponding changes and refinements made in the revised paper are summarized in our response below. The actual comments and questions of the reviewer are in BOLDED RC, and the author's responses are italicized AR.

Reviewer

RC1. Abstract: the information why biostimulants are so important should be added. On the other side, the results are to descriptive in this part.

AR/ Thank you for your comment. The information of biostimulants has been added to the abstract section

RC2. Authors did not make a research hypothesis. However, the aims of the study are clearly stated at the end of the introduction section.  

AR/ Thank you for your comment. The hypothesis has been added to the revised manuscript

RC3. MM section: the information about conditions during experiments should be described: what type of soil was used, by which tool the moisture of soils was measured, what was the temperature and air humidity?

AR/ Thank you for your comment. This information is presented in MM in the revised manuscript

RC4. MM section: the statistical procedure should be described more detailed: what was p value, what was the number of repetitions? Moreover, in figure and table captions should be explained what symbols ± means (SE or SD?).

AR/ Thank you for your comment. All information about the statistical analysis has been inserted and modified in the revised version.

RC5. In conclusion there should mechanistic response/ description what the investigated mechanism looks like. 

AR/ Thank you for your comment. All information about the most important results has been presented in conclusion

Best regards

Author Response

Manuscript number: agriculture- 2072747

Paper title: Effects of Oak Leaf Extract, Biofertilizer, and Soil Containing Oak Leaf Powder on Tomato Growth and Biochemical Characteristics Under Water Stress Conditions

Authors: Nawroz Abdul-razzak Tahir, Kamaran Salh Rasul, Djshwar Dhahir Lateef, Florian M. W. Grundler

Dear Editor and Reviewer

The authors would like to thank the area editor and the reviewers for their precious time and invaluable comments. We have carefully addressed all the comments. The constructive comments/ suggestions by the reviewer are really appreciated. We have now completely revised the manuscript. All corrections in English language and updating of information are highlighted by the red lines. The corresponding changes and refinements made in the revised paper are summarized in our response below. The actual comments and questions of the reviewer are in BOLDED RC, and the author's responses are italicized AR.

Reviewer-4

RC1. However, the captions in Tables and Figures should be amended.

AR/ Thank you. The captions have been modified and improved

RC2. In addition, English is decent but I suggest a thorough review of the manuscript before accepting it for publication.

AR/ Thank you. The spelling and grammar have been checked and corrected

RC3. Abstract; Make the title a simple statement, Give the problem statement in a single line, give a reason for the selection of the current technique, Quantitative data is also important to support your conclusion, would you please provide some quantitative data in terms of percentage significant increase or decrease in the abstract, please provide a conclusive conclusion with is withdrawn through research in a single line, Give future prospective in a single line.

AR/ Thank you. The abstract has been modified

RC4. As per standard suggestions, please avoid using title words as keywords.

AR/ Thank you. The words used for forming the title are not used as the keywords

RC5. 8.  Please follow the title and rewrite the introduction in the following sequence as i.e., use of complementary light, photosynthetic,  Paros in salinity and alkaline stress , problem statement, aims of study and hypothesis.

AR/ Thank you. The introduction section has been modified.

RC6. Also, provide a novelty statement at the end. What new things authors have done or correlated in this research compared to old ones?

AR/ Thank you. This section has been modified

Would you please give a single line about the knowledge gap which your research has covered along with the hypothesis statement?

AR/ Thank you. The hypothesis statement has been added to the revised version.

RC7. Materials and methods: Please provide a reference for statistical analysis.

AR/ The references have been added

RC8. Results and Discussion. Very descriptive. Please give only significant results. Also, give mechanistic discussion. It is not a correct way to discuss results based on other scientists' findings. Please elaborate on specified mechanisms which are regulating and result. Please rewrite the results and discussion again

AR/ Thank you. The results and discussion are well written and the mechanism of effect of SOES, SOS, and SOBS has been discussed in the manuscript. Some modification has been modified in this section in the revised manuscript.

RC9. Conclusion: add the targeted beneficiary audience who will get benefits from this research. Also, give clear-cut recommendations

AR/ Thank you. The target and recommendation are presented in the conclusion section.

RC10. 14 In spite this is research article a lack of recent literature (Recent references (last 3 years), therefore the authors should include the most recent references on this subject

AR/ Thank you. Some recent references have been added to the revised manuscript. Due to the lacking of researches about the using of oak tissues as biostimulator, we are obligated to use these references

RC11. Standardize references

AR/ Thank you. All references are standardized

Best regards

Reviewer 3 Report

The manuscript "Effects of Oak Leaf Extract, Biofertilizer, and Soil Containing Oak Leaf Powder on Tomato Growth and Biochemical Characteristics Under Water Stress Conditions" is well written and discussed. Authors have done a good job on the manuscript. I only suggest that they look at a few points that i've raised and consider having the manuscript title page on the supplementary information too.

Author Response

Manuscript number: agriculture- 2072747

Paper title: Effects of Oak Leaf Extract, Biofertilizer, and Soil Containing Oak Leaf Powder on Tomato Growth and Biochemical Characteristics Under Water Stress Conditions

Authors: Nawroz Abdul-razzak Tahir, Kamaran Salh Rasul, Djshwar Dhahir Lateef, Florian M. W. Grundler

Dear Editor and Reviewer

The authors would like to thank the area editor and the reviewers for their precious time and invaluable comments. We have carefully addressed all the comments. The constructive comments/ suggestions by the reviewer are really appreciated. We have now completely revised the manuscript. All corrections in English language and updating of information are highlighted by the red lines. The corresponding changes and refinements made in the revised paper are summarized in our response below. The actual comments and questions of the reviewer are in BOLDED RC, and the author's responses are italicized AR.

Reviewer

RC1. L54: which areas?

AR/ Thank you for your comment. The area represents the world. This line has been modified

RC2. L68: This sentence is lost here.

AR/ Thank you for your comment. This line has been deleted

RC3. L96: This is not true.

AR/ Thank you for your comment. This line has been modified

Best regards

Reviewer 4 Report

·         Check MS for typographic errors and grammatical mistakes.

·         Add a picture of tomato plants showing effect of drought and other treatments and separate picture of tomato fruits at the time of harvesting.

·         A figure showing experimental set up and treatments will be helpful for the readers.

·         Justify the selection of oak leaves for the study. Add about oak leaves in introduction. Cite appropriate references.

·         Treatment of biofertilizer is all together different. I think there should be one more treatment i.e. stressed plants + biofertilizer alone

·         Please mention source of tomato seeds in materials.

·         Please justify selection of oak leaf powder and oak leaf extract both. Here also, there must be one more treatment i.e. stressed plants + oak leaf extract

·         Describe about collected leaves of oak i.e. their size, shape, physiological stage, age etc.

·         Is there any reference for preparation and application for oak leaf extract?

·         The drought treatment is confusing. How the first and second stress is combined? Please write in detail.

·         Mention the size of plastic pots used.

·         Mention duration of stress treatment at the end of -- 2.2. Experimental Design Components, Plant Treatment, and Growth Conditions

·         Mention the tissue used for Lipid Peroxidation Assays. fruit or leaf powder?

·         Why there are two references for estimation of MDA content? Also check reference style for second reference.

·         Cite reference for calculation of trait index.

·         Mention the exact time of oak leaf powder/extract treatment at pre-flowering and pre-fruiting stage and their combination.

·         Data representation can be improved. For example, instead of table on Substances detected by GC/MS analysis in the leaf water extract of Quercus Aegilops—a heat map will be more informative and easier to understand. OR in table add one more column of bioactivity of identified metabolites of oak lead extract. It will be helpful to readers to understand their role in drought stress.

·         In supplementary data, give actual data values of all the experiments.

Author Response

Manuscript number: agriculture- 2072747

Paper title: Effects of Oak Leaf Extract, Biofertilizer, and Soil Containing Oak Leaf Powder on Tomato Growth and Biochemical Characteristics Under Water Stress Conditions

Authors: Nawroz Abdul-razzak Tahir, Kamaran Salh Rasul, Djshwar Dhahir Lateef, Florian M. W. Grundler

Dear Editor and Reviewer

The authors would like to thank the area editor and the reviewers for their precious time and invaluable comments. We have carefully addressed all the comments. The constructive comments/ suggestions by the reviewer are really appreciated. We have now completely revised the manuscript. All corrections in English language and updating of information are highlighted by the red lines. The corresponding changes and refinements made in the revised paper are summarized in our response below. The actual comments and questions of the reviewer are in BOLDED RC, and the author's responses are italicized AR.

Reviewer

RC1. Check MS for typographic errors and grammatical mistakes.

AR/ Thank you for your comment. The typographic errors and grammatical mistakes have been checked and corrected in the revised version

RC2. Add a picture of tomato plants showing effect of drought and other treatments and separate picture of tomato fruits at the time of harvesting.

AR/ Thank you for your comment. A Figure S2 has been added to the revised manuscript as supplementary data in the supplementary file.

RC3. A figure showing experimental set up and treatments will be helpful for the readers.

AR/ Thank you for your comment. A Figure S1 has been added to the revised manuscript as supplementary data in the supplementary file.

RC4. Justify the selection of oak leaves for the study. Add about oak leaves in introduction. Cite appropriate references.

AR/ Thank you for your comment. The information about the oak leaves with citation have been added to the introduction

RC5. Treatment of biofertilizer is all together different. I think there should be one more treatment i.e. stressed plants + biofertilizer alone

AR/ Thank you for your comment. We did not have a treatment with stressed plants + biofertilizer in our experiment because the role of biofertilizer is already known as mentioned in the material, and methods, and discussion. In addition, we added this biofertilizer to the soil because it contains the rhizosphere bacteria and some enzymes that help and facilitate the degrading of leaf powder.

RC6. Please mention source of tomato seeds in materials.

AR/ All tomato genotypes were collected from the Agricultural Research Center of the Ministry of Agriculture and Water Resources in Kurdistan, Iraq.

RC7. Please justify selection of oak leaf powder and oak leaf extract both. Here also, there must be one more treatment i.e. stressed plants + oak leaf extract

AR/ Thank you for your comment. We did not have a treatment with stressed plants + oak leaf extract in our experiment.

RC8. Describe about collected leaves of oak i.e. their size, shape, physiological stage, age etc.

AR/ Thank you for your comment. Fully developed and healthy oak leaves (Quercus aegilops Oliv.) were gathered at the vegetative stage on May 17, 2021.

RC9.  Is there any reference for preparation and application for oak leaf extract?

AR/ Thank you for your comment. The references have been added

RC10. The drought treatment is confusing. How the first and second stress is combined? Please write in detail.

AR/ Thank you for your comment. The experiment is divided into three groups: The plants in Group 1 were stressed before flowering. The plants in the second group were stressed prior to fruiting. The third category includes plants that were stressed before flowering and fruiting.

RC11. Mention the size of plastic pots used.

AR/ Thank you for your comment. The height and diameter of plastic pot are 40 cm and 18 cm, respectively. This information has been added to the revised manuscript

RC12. Mention duration of stress treatment at the end of -- 2.2. Experimental Design Components, Plant Treatment, and Growth Conditions

AR/ Thank you for your comment. The duration of stress for each stage has been added.

RC13.  Mention the tissue used for Lipid Peroxidation Assays. fruit or leaf powder?

  1. Thank you for your comment. The leaves powder is used for lipid peroxidation assay. This information has been added to the revised manuscript

RC14. Why there are two references for estimation of MDA content? Also check reference style for second reference.

AR/ Thank you for your comment. The second reference has been deleted

RC15. Cite reference for calculation of trait index.

AR/ Thank you for your comment. A reference has been added in the revised version of the manuscript

RC16.  Mention the exact time of oak leaf powder/extract treatment at pre-flowering and pre-fruiting stage and their combination.

AR/ Thank you for your comment. As mention in materials and methods, the oak leaf powder has been added or mixed with the soil before planting of tomato plants. The extract is applied four times by foliar spray before flowering (first stress stage), and fruiting (second stress stage) with three-days intervals. Leaf extract is sprayed before flowering on June 7, June 10, June 13, and June 16, 2021, and before fruiting on July 15, July 18, July 21, and July 24, 2021.

RC17. Data representation can be improved. For example, instead of table on Substances detected by GC/MS analysis in the leaf water extract of Quercus Aegilops—a heat map will be more informative and easier to understand. OR in table add one more column of bioactivity of identified metabolites of oak lead extract. It will be helpful to readers to understand their role in drought stress.

AR/ Thank you for your comment. The biological activity of major compounds has been added to Table 7

AR18. In supplementary data, give actual data values of all the experiments.

AR/ Thank you for your comment. All data in supplementary file represent the actual data, which represent by the mean ± standard deviation (SD. Each value is the average of 8 plants or replications.

Best regards

Round 2

Reviewer 2 Report

Authors tracked every comment very good so now the paper should be accepted in present form

Reviewer 4 Report

Authors have addressed all the comments.